# Molecular Survey of *Bartonella* Species in Stray Cats and Dogs, Humans, and Questing Ticks from Portugal

**DOI:** 10.3390/pathogens11070749

**Published:** 2022-06-30

**Authors:** Estefania Torrejón, Gustavo Seron Sanches, Leonardo Moerbeck, Lenira Santos, Marcos Rogério André, Ana Domingos, Sandra Antunes

**Affiliations:** 1Instituto de Higiene e Medicina Tropical, Universidade Nova de Lisboa (IHMT-UNL), Rua da Junqueira 100, 1349-008 Lisbon, Portugal; tefi1805@hotmail.com (E.T.); gustavoseron@hotmail.com (G.S.S.); moerbeck.leonardo@gmail.com (L.M.); lenira_santos@hotmail.com (L.S.); santunes@ihmt.unl.pt (S.A.); 2Escola de Ciências da Vida, Pontifícia Universidade Católica do Paraná, Rua Imaculada Conceição 1155, Curitiba 80215-901, PR, Brazil; 3Global Health and Tropical Medicine, Instituto de Higiene e Medicina Tropical, Universidade Nova de Lisboa (GHTM-IHMT-UNL), Rua da Junqueira 100, 1349-008 Lisbon, Portugal; 4Departamento de Patologia Veterinária, Universidade Estadual Paulista (FCAV-UNESP), Via de Acesso Paulo Donato Castellane, s/n, Jaboticabal 14884-900, SP, Brazil; marcosandre.fcav@gmail.com

**Keywords:** *Bartonella clarridgeiae*, *Bartonella henselae*, cat scratch disease, Portugal, ticks, hosts

## Abstract

*Bartonella* spp. comprises emergent and re-emergent fastidious Gram-negative bacteria with worldwide distribution. Cats are the main reservoir hosts for *Bartonella henselae* and dogs represent opportunistic hosts for the bacteria. Even though ticks may also play a role in transmission, their competence as vectors for *Bartonella* spp. has not been totally understood. Considering only a few studies had a focus on screening *Bartonella* in animals, humans and ectoparasites in Portugal, this study aimed to address the molecular occurrence of *Bartonella* sp. in 123 stray cats, 25 stray dogs, 30 humans from Lisbon and 236 questing ticks within the country. Using a qPCR targeting the *nuoG* gene, it was possible to detect *Bartonella* sp. DNA on 20.32% of cat samples (25/123). From these positive samples, 13 sequences were characterized as *B. henselae*, 11 as *B. clarridgeiae* and 1 presented co-infection with both species. The absolute quantification of *nuoG*
*Bartonella* DNA in sampled cats ranged from 2.78 × 10 to 1.03 × 10^5^ copies/µL. The sampled dogs, humans and ticks were negative. These results showed that *B. henselae* and *B. clarridgeiae* are circulating in stray cats from Lisbon. Additional and more extended studies should be conducted to determine the impact of such infections on humans, particularly those in constant and direct contact with cats.

## 1. Introduction

*Bartonella* spp. encompasses small, fastidious, and facultative Gram-negative intracellular bacteria with a global distribution [1] belonging to alpha-2 class of Proteobacteria, order Rhizobiales, and family Bartonellaceae [2]. Currently, there are over 45 species described, which have been isolated from humans, domestic and wild animals [3]; 15 out of these species are known to be associated with human infections [1]. Transmission by sand flies, fleas, and lice has been properly documented while the role of ticks is still under discussion [4,5]. The existence of non-vectorial transmission routes has also been evaluated, as some reports have been made describing transmission via scratch and bite of infected cats, the use of infected needles, and blood transfusion [6,7,8]. *Bartonella* spp. are neglected zoonotic pathogens that infect mostly erythrocytes and endothelial cells, in which they can survive for a prolonged period, resulting in a long-lasting infection, usually linked to a relapsing pattern of bacteremia [9].

*B. henselae* is one of the species that has been frequently identified in humans, as well as in companion animals, especially cats [1]. Other *Bartonella* spp. with relevant zoonotic importance are *B. clarridgeiae* and *B. washoensis*, and both these species have been previously isolated from dogs [10]; moreover, all *Bartonella* spp. identified in sick dogs are known as pathogenic or potentially pathogenic infectious agents for humans, suggesting that dogs might act as useful sentinel species and important comparative models for human infections [11]. It is noteworthy that the vast majority of infections in humans transmitted by arthropod vectors have been attributed to *Bartonella quintana* and *Bartonella bacilliformis*, but only *B. henselae* has been reported as a result of mechanical infection due to contact with an animal reservoir (cat scratch or bite) [12]. Many studies have been carried out to identify *Bartonella* sp. in ticks due to a growing interest in investigating their role as potential vectors for these bacteria [13]. Despite some evidence of the vectorial competence of ticks for bartonellae through experimental studies using artificial feeding, their vectorial capacity for bartonellae has not been confirmed yet. An integrated study involving hosts, reservoirs, and vectors for *Bartonella* sp. would be the best approach to evaluate the current prevalence in each population and select the best control measures to be implemented [14].

To date, data associated with *Bartonella* infection in humans from Europe are scarce, being more common in domestic cats and dogs [15]. For Portugal, molecular surveys have been performed on cats: while in 1995, 49% (25/51) of cats were found positive by a conventional PCR targeting the citrate synthase gene (*gltA*) [16], in 2014 an occurrence of 2.9% (19/649) for *Bartonella* spp. was reported among cats using a qPCR targeting the intergenic region 16S-23S rRNA (*ITS*) [17]; however, no molecular surveys have been performed in humans, dogs or ticks, demonstrating that more studies need to be performed to determine the current prevalence of *Bartonella* sp. in other hosts besides cats (the main reservoirs for *B. henselae*, *B. clarridgeiae* and *B. koehlerae*) and in potential vectors, such as ticks.

The present research work aimed to survey *Bartonella* sp. DNA presence in cats, dogs, humans from the urban area of Lisbon and ticks from Portugal mainland, using a qPCR assay based on the *nuoG* gene; moreover, a characterization of the *Bartonella* sp. detected was also carried out using *gltA* and *ribC*-based cPCR assays, in order to identify the *Bartonella* species circulating in Lisbon, Portugal.

## 2. Results

### 2.1. Tick Samples and Molecular Detection of Bartonella spp.

A total of 268 specimens were used in the present study originally collected in 19 geographical points from eight districts of mainland Portugal from 2012 to 2018 (Figure 1). Of these, 114 (42.53%) were identified as *Ixodes ricinus*, 18 (6.71%) as *Ixodes* sp. and 136 (50.75%) as *Rhipicephalus sanguineus* sensu lato. For *I. ricinus*, 38 (33.3%) specimens were female, 56 (49.12%) were male and 20 (17.54%) nymphs; in addition, all the 18 (100%) individuals classified as *Ixodes* sp. were nymphs. Regarding *R. sanguineus*, 68 (50%) individuals were females and 68 (50%) males. Integrity of all samples was confirmed by PCR targeting the tick 18S rRNA gene. The presence of *Bartonella* spp. DNA was not detected in any of the samples.

### 2.2. Detection of Bartonella spp. in Human Blood and Blood from Non-Domiciliated Cats and Dogs

The number of samples obtained for each group namely, cats, dogs and humans were 123, 25 and 30, respectively. Information about age, sex and presence of fleas is provided in Table 1, excluding humans’ samples, from which no information is available. Samples were positive for the first screening PCR step (targeting GAPDH), confirming DNA integrity for all samples.

Concerning the molecular detection of *Bartonella* sp., 25 (20.3%) of 123 cat samples were positive for the qPCR targeting *nuoG*. The rest of the samples, corresponding to ticks, dogs, and humans (including blood cultures done for dogs and humans) were negative for the presence of *Bartonella* sp. Reaction parameters of the qPCR assays for each type of sample are listed in Table 2.

Out of the 123 blood samples, 25 (20.33%) were positive for at least one *Bartonella* spp. Of these 25 positive cats, 13 were females (17.1% of all female cats) and 12 were males (23.52% of all males). Of the 11 juvenile cats, 5 were infected (45.45%), as well as 20 (17.85%) of the total adults. Only three of the nine cats (33.33%) presenting flea infestation were found to be positive for *Bartonella* spp. and all three were females. In addition, 28% of these 25 cats were rescued from very close areas (Alvalade, Misericórdia and Campolide).

All the 25 cat blood samples, positive in the qPCR, targeting the *nuoG* gene, also amplified the targeted fragment of the *gltA* gene of *Bartonella* by cPCR. The BLAST analysis showed that 13 out of the 25 samples (52%) presented an identity ranging from 99% to 100% with *B. henselae* and 11 (44%) showed an identity ranging from 99% to 100% with *B. clarridgeiae*. Regarding the *ribC* gene, 24 samples were found to be positive and confirmed *gltA* results. One of the samples presented heterozygous (double) peaks, therefore it went through molecular cloning and was confirmed to harbour a co-infection with both *B. henselae* and *B. clarridgeiae* and 14 of the 25 positive sequences (56%) presented identity ranging from 99.61% to 100% with *B. henselae* and 11 (44%) showed an identity ranging from 99.81% to 100% with *B. clarridgeiae*. The *gltA* and *ribC* sequences obtained from positive cats were deposited in the GenBank international database under the following accession numbers: MN564828 and MN564829 for *gltA*; MN564824, MN564825, MN564826, MN564827 and MN632451 for *ribC*.

For the dataset of *B. henselae gltA* (13 sequences), the number of sites found was 717, Pi = 0.000, Hd = 0.000 and only one haplotype was identified (MN564828). In the same way, the dataset of *B. clarridgeiae gltA* (11 sequences) reported a number of sites of 743, Pi = 0.000, Hd = 0.000 and only one haplotype was identified (MN564829). Furthermore, for the dataset of *B. henselae ribC* (14 sequences), the number of sites found was 530, Pi = 0.00081, Hd = 0.2747 and three haplotypes were identified: haplotype 1 (Hap1, MN564824), corresponding to 12 samples (2, 16, 27, 29, 37, 57, 62, 86, 90, 107, 108, and 119); haplotype 2 (Hap2 MN564825), corresponding to sample 95, and haplotype 3 (Hap3, MN564826), corresponding to sample 101. Similarly, regarding the *B. clarridgeiae ribC* group (11 sequences), the number of sites found were 521, Pi = 0.00035, Hd = 0.182 and two haplotypes were identified: Haplotype 1c (Hap1c, MN564827), corresponding to 10 samples (19, 22, 28, 33, 46, 52, 79, 80, 97, and 118), and Haplotype 2c, corresponding to sample 108 (Hap2c, MN632451).

From the *gltA* sequences, and specifically from the 13 ones characterized as *B. henselae*, eight (61.53%) showed a 99% identity with *B. henselae* strain Houston-I (Accession number: CP020742) and five (38.46%) with an identity ranging from 99% to 100% with *B. henselae* isolated from cat fleas from Chile (Accession number: KY913625). Out of the 11 sequences identified as *B. clarridgeiae*, eight (72.72%) presented an identity of 99% with *B. clarridgeiae* strain 73 (Accession number: FN645454), two (18.18%) a 100% identity with isolation from cat fleas from Chile (Accession number: KY913629), and one (9.09%) a 100% identity with another isolation from Chilean cat fleas (Accession number: KY913636); one sequence was not analyzed due to the presence of two overlapping peaks and a failed molecular cloning procedure.

Regarding the *ribC* sequences, out of the 14 samples corresponding to a *B. henselae* characterization, 8 (61.53%) had a range of 99.61% to 99.81% of identity with isolation from mammal fleas in South Carolina, USA (Accession number: AY953284), 5 (38.46%) had an identity ranging from 92.62% to 100% with *B. henselae* strain Houston-I (Accession number: CP020742), and 1 (7.69%), showed 100% of identity with isolation from cats in Brazil (Accession number: HM588661); moreover, from the group characterized as *B. clarridgeiae*, all 11 sequences (100%) presented an identity ranging from 99.81% to 100% with an isolation from cats in southern Brazil (Accession number: KR092386).

The Maximum Likelihood phylogenetic tree built for the *gltA* gene (showed in Figure 2) grouped the unique haplotype for *B. henselae* (Accession number: MN564828) in the same cluster with other *B. henselae* from different countries and reservoirs, including one isolated from cats, in Brazil (Accession number: MH019304.1), and cat fleas, in Austria (Accession number: MF374384.1), supported by a bootstrap value of 99%. In addition, the unique haplotype for *B. clarridgeiae* (Accession number: MN564829) was grouped in a cluster together with other *B. clarridgeiae* strains, including two isolations from cats in Brazil (Accession number: MH019302.1) and Thailand (Accession number: KX001761.1) and one from cat fleas from Chile (Accession number: KY913636.1), supported by a 100% bootstrap value. Additionally, the *ribC* Maximum Likelihood phylogenetic tree (Figure 3) grouped all three haplotypes for *B. henselae* (Accession numbers: MN564824, MN564825 and MN564825) on the same clade, together with sequences corresponding to isolations from cats in Brazil (Accession number: HM588661) and fleas in the USA (Accession number: AY953284); this cluster had a 100% of bootstrap support value, and moreover, the two haplotypes for *B. clarridgeiae* (Accession numbers: MN564827 and MN632451) formed a cluster supported for a 100% of bootstrap value, together with isolations from cats in Brazil (Accession number: KR092386), Japan (Accession number: AB292604) and China (Accession number: EU571943).

## 3. Discussion

In this study, 20.32% of the sampled cats were positive for *Bartonella* sp. On the other hand, all blood samples from dogs and humans and questing ticks were negative for bartonellae DNA, even though the screening was also performed in liquid and solid cultures (for dogs and humans), to increase the sensitivity of the detection assay.

The occurrence of *Bartonella* spp. among cats in Lisbon was higher (20.3%) than the one found in cats from southern Portugal [17], where only 2.9% of cats were positive for *Bartonella* sp; this finding could be explained by the fact that in the latter study, domestic cats constituted half of the studied samples. Domiciled cats typically are at lesser risk of flea infestation when compared to stray cats, which lack ectoparasiticide treatment [18,19]. In regard to the study carried out mainly in Évora, where 49% of the screened samples were found positive [16], the high prevalence could be explained by the fact that said city is in the countryside, presenting conditions that are more favorable for vector infestation in animals. Additionally, the molecular method selected for both studies was based on a cPCR, whereas the present study used a qPCR assay, known for being more specific and sensitive. In the present study, we found a higher percentage of juvenile cats infected with the bacteria in comparison to adult cats. It has been demonstrated that cat age is a risk factor for *Bartonella* spp. infection, a feature that can be associated with a more immature immune system. As a result, younger cats are more prone to present peaks of parasitaemia, which ultimately can facilitate pathogen detection [19,20]. It is important to denote that the combination of *Bartonella* alphaproteobacteria growth medium (BAPGM) followed by qPCR and chocolate agar culture plus the new qPCR round was not employed in the cat blood samples screened which would increase sensitivity and possibly yield in a higher number of positive samples [21].

Based on the phylogenetic analysis, for both *gltA* and *ribC* sequences of *B. henselae* and *B. clarridgeiae*, the bacteria identified in the present study, were found to be close to other *Bartonella* isolates, not only from cats, but from other mammals and vectors as well. *B. henselae* genotypes were closely related to isolates from cats in Brazil [22], dogs in Chile [23], cat fleas in Austria [24] and humans in Australia [25]. In a similar way, *B. clarridgeiae* sequences were related to isolates from cats in Brazil [22], cats in Thailand [26] and cat fleas from Chile [23]. These results could suggest that strains of both *B. henselae* and *B. clarridgeiae* are being shared by animals from different locations, possibly due to human and animal migration. The genetic diversity of *B. henselae* and *B. clarridgeiae* circulating in cats from Lisbon and other Portuguese regions can and should be addressed in the future to better understand disease epidemiology. Multilocus sequence typing (MLST) studies have been conducted on *Bartonella* spp. contributing to clarifying phylogenetic relationships, host specificity, diversity and the discovery of novel species [21,27,28,29].

In regard to dogs, despite the high prevalence detected for *Bartonella* sp. in cats from Lisbon, all dog samples were qPCR negative. It is widely accepted that dogs are susceptible to infection, but their role as reservoirs is not clear and studies suggest that dogs are incidental hosts for *Bartonella* spp. [15,30]. In dogs, infections often result in clinical manifestations similar to those seen in humans, and endocarditis is the most observed disorder. Apparently, in Europe there is a lower prevalence of infection compared to other continents [15,31], to which vector abundance, prevention measures and even socio-demographic differences may be contributing factors; however, the lack of a gold standard diagnostic test may be the greater limitation here. A recent study has pointed out the discrepancies on accuracy and sensitivity between different methodologies, including digital PCR (ddPCR) [32], showing that direct and absolute comparisons are often impossible. Many of the studies targeting dogs are serology based [17,31], while detection in blood by PCR or qPCR has been rarely reported [15]. There are several reports showing PCR failure to amplify the bacteria from the blood of infected individuals, and even ddPCR did not show the needed sensitivity (reviewed in Ref. [32]). Tissue samples provide more robust results; however, these types of samples are less available (usually taken from individuals presenting disease manifestations or very specific studies) making blood a preferred sample for screening purposes. Taking these limitations into account, well-designed PCR assays should support qPCR detection with PCR after BAPGM enrichment culture [33,34]. Blood cultures can be used to amplify bacteria and unmask the infection, a strategy that was conducted in the present study and resulted in the confirmation of negative infections. A previous survey carried out in Spain [35], which applied a qPCR technique targeting the *ITS* gene, had a sample of dogs previously diagnosed with endocarditis, while the present study sampled healthy or asymptomatic dogs, making it less likely to find individuals infected with *Bartonella* sp. Nevertheless, considering that the two species of *Bartonella* detected in cats (*B. henselae* and *B. clarridgeiae*) had also been reported in dogs in previous studies [10,36,37,38,39], more surveys need to be performed in Lisbon, selecting preferably a more diverse group of dogs, in terms of their medical condition.

Despite *Bartonella* sp. infection rates in humans show a tendency to be higher in immune-compromised people [40,41], some species are associated with other specific risk factors. For *B. henselae* and *B. clarridgeiae*, infection in humans is strictly associated with contact with cats and exposure to cat fleas [42]. For this reason, the sampled group of human individuals consisted of veterinary personnel and volunteers of a cat-rescue association, who were in constant contact with the felines. The limitations of these screenings in humans are similar to those in dogs. In Europe, *Bartonella* sp. has been previously detected in veterinary practitioners by a cPCR method targeting the *ITS* gene [43] and on cat owners from Spain (using the same method) [30], with a prevalence of 28% and 27%, respectively. For serology assessments, an incidence of 7% was reported for *B. henselae* and 1% for *B. quintana* on a survey performed in Sweden, with a sample of 224 patients [44], while another survey performed in Poland had a reported incidence of 23% seropositive cases for *B. henselae* and 2.85% for *B. quintana* from 105 tested subjects, composed of 65 blood donors and 40 patients suffering from musculoskeletal symptoms [45]. Despite these reports, the samples screened in the present study were negative for the bacteria, also including the blood cultured samples. In Portugal, few cases have been reported, which would suggest *Bartonella* sp. infections are not common. A report by the National Health Institute (INSA by its Portuguese acronym) from 2012, indicates a rate of 17% of seroprevalence of *Bartonella* infections in Portugal, from a total of 189 suspected cases [46], but taking into account the whole Portuguese population (according to the national statistical institute, INE, in 2021, the Portuguese population was estimated in 10 344 802 individuals, https://censos.ine.pt (accessed on 23 June 2022)), these numbers still suggest a low incidence. In addition, most of the previous reports [47,48] were on patients that had a pre-existing condition that could have made them more vulnerable to the bacteria, with the exception of a recent report on a previously healthy adolescent [49]. Furthermore, there is a report of positive serology for *Bartonella* sp., in which the blood culture gave a negative result [47], adding to the need to implement a standard gold method for the detection of *Bartonella* spp. in hosts other than cats. Consequently, more surveys need to be developed in the country to clarify the impact of bartonellosis on humans.

In the last few years, many efforts have been made to clarify the role of ticks in the transmission of *Bartonella* spp. The extraordinary vector capacity of these ectoparasites together with reports of the presence of *Bartonella* spp. DNA in ticks and, more recently, the reports of *Bartonella* spp. seroprevalence in tick-exposed patients from Sweden and Poland, although the last one lacked statistical significance [44,45], which has put them on the spot [50]. In the present study, despite sampling over 260 ticks collected in different locations across Portugal mainland, *Bartonella* sp. was not detected in any of the samples. More than half of the screened ticks were classified as *Rhipicephalus sanguineus* sensu lato and, 42.53%, as *Ixodes Ricinus*; these species were chosen due to their great importance for public health, with *I. ricinus* being the most medically important tick in Portugal (and Europe) and *R. sanguineus* being the most prevalent tick in companion animals [51]. In 2010, Angelakis et al. published a review [13] that included a series of molecular surveys for *Bartonella* carried out in ticks from 1995 to 2010, in which a diversity of *Bartonella* spp. were detected from different tick species (*Amblyomma americanum*, *Carios kelleyi*, *Dermacentor occidentalis*, *D. reticulatus*, *D. variabilis*, *Haemaphysalis flava*, *H. longicornis*, *Ixodes nipponensis*, *I. pacificus*, *I*. *persulcatus*, *I. ricinus*, *I. scapularis*, *I. sinensis*, *I. turdus*, *R. sanguineus*), from numerous geographic locations. Since then, more studies have been published, some reporting the pathogen in ticks in a given region [52,53], others, like the present study, the absence of the pathogen on ticks recovered from hosts [54]. Residual pathogen DNA, acquired during a previous blood meal, can be detected after tick molt and also *Bartonella* DNA within a tick does not imply that the tick might be able to transmit the bacteria during the course of blood-feeding, therefore not conferring epidemiologic risk [55]. The generalist tick *I. ricinus*, has often been used in studies performed to clarify its competence to transmit the bacteria. The transovarial and transstadial transmission of *B. henselae*, *B. grahamii*, and *B. schoenbuchensis* have been recently demonstrated [56], while the transmission of *B. birtlesii* to susceptible mice, using laboratory-infected *I. ricinus*, was previously achieved [57]. *Rhipicephalus sanguineus* being the most prevalent tick in companion animals, has recently been used in similar studies [58,59] but evidence that this tick species can be the vector of *Bartonella* sp. is also still missing. Altogether, and despite the efforts, more studies are required to clearly establish if ticks are efficient vectors of *Bartonella* or rather a more opportunistic and weaker threat.

## 4. Material and Methods

### 4.1. Ethics Statement

The Institute of Hygiene and Tropical Medicine Ethics Council approved this study under process number 01.19. Animal experiments were conducted according to the principle of the 3Rs, to replace, reduce and refine the use of animals for scientific purposes. Humans agreed to participate in the research study and signed an informed consent term.

### 4.2. Biological Samples Collection

#### 4.2.1. Cats and Dogs Whole Blood Sampling

A total of 123 non-domiciliated cats and 30 non-domiciliated dogs participating in the sterilization program promoted by “Casa dos Animais de Lisboa” (CAL) were sampled from January to July 2019. After anaesthesia and immediately before the sterilization procedure, a volume of 400 µL of blood was collected from the jugular vein of each cat, and from the radial vein of each dog, and aliquoted into two EDTA-containing tubes. One tube was kept under refrigeration at 4 °C until DNA extraction and the other were promptly frozen at −80 °C until use for *Bartonella* isolation. For cats, culture was not performed. Information about gender, breed, age, and presence of ectoparasites was registered for each animal.

#### 4.2.2. Humans Whole Blood Sampling

Thirty healthy individuals frequently exposed to bites or cat scratches, (veterinarians, veterinary technicians, volunteers of associations for the protection of homeless animals and cat owners) participated in this study, from February to July 2019. Approximately 1 mL of EDTA whole blood was collected from each volunteer’s median cubital vein, after the signing the informed consent term. The collected whole blood was divided into two aliquots, one tube was kept under refrigeration at 4 °C until DNA extraction for a direct qPCR and the other was frozen at −80 °C until use for *Bartonella* isolation.

#### 4.2.3. Ticks

The ticks used in the present study were obtained and processed previously under past institutional research projects. The tick collection was obtained from 2012 to 2019, covering 19 geographical points of mainland Portugal. Briefly, ticks were collected by dragging-flagging vegetation [60], examined under a stereomicroscope, and morphologically identified using taxonomic keys [61,62]. Each specimen was placed in a 1.5 mL labelled microtube and stored in 70% ethanol at −20 °C, until DNA extraction.

### 4.3. Cultivation of Bartonella from Dog and Human Blood Samples

The EDTA-whole blood samples were thawed at room temperature and 100 µL were collected and mixed with Schneider’s insect medium (Sigma-Aldrich, Inc., St. Louis, MO, USA) supplemented with 10% of bovine calf serum (Biological Industries, Kibbutz Beit Haemek, Israel), 5% of sucrose (VWR International, Ltd., Poole, UK) and amphotericin B (Biological Industries, Kibbutz Beit Haemek, Israel) to a final concentration of 2.5 µg/mL [63,64]. For liquid cultures, 200 μL of blood sample was added to 2 mL of medium and for solid cultures, 200 μL of the homogenized solution was diluted 1:1 in a liquid medium and seeded onto chocolate agar plates (Novamed, Ltd., Jerusalem, Israel), and incubated at 37 °C in a 5% CO_2_ atmosphere. The plates were screened for the growth of suggestive *Bartonella*-like colonies (i.e., slow-growing, creamy or dry rounded colonies) after 3 days post-incubation, and then every 48 h, for 42 days [65].

### 4.4. DNA Extraction

Extraction of DNA from ticks was performed using alkaline hydrolysis, following a previously described protocol [66]. Briefly, specimens were boiled in 0.7 M ammonium hydroxide to free the DNA. After the lysate was air-dried to evaporate the ammonia. Extraction of cat, dog and human DNA was made from 200 µL of whole blood EDTA was performed using the NZY Blood gDNA Isolation Kit (NZYTech, Lisbon, Portugal), following the manufacturer’s recommendations. The same method for DNA extraction was used for isolated and resuspended colonies consistent with *Bartonella* sp. in 100 µL of ultra-pure water. All purified DNA samples were eluted in 30 µL of elution buffer (NZYTech).

### 4.5. Polymerase Chain Reaction (PCR)

#### 4.5.1. PCR Inhibitors and DNA Integrity

The ratio of absorbance at 260 nm and 280 nm was used to assess the purity of DNA on a NanoDrop ND-1000 spectrophotometer (Thermo Scientific, Wilmington, NC, USA). The DNA extracted from cats’, dogs’ and humans’ blood samples, and also from the liquid cultures was used on a conventional PCR (cPCR) targeting a 400 base pairs (bp) fragment of the mammalian glyceraldehyde-3-phosphate dehydrogenase protein family gene (*gapdh*) to evaluate PCR inhibition [67]. Ticks’ DNA was used for the same purpose in a cPCR targeting a 653-bp fragment of the gene coding for the small subunit ribosomal RNA of arthropods (18S rRNA), according to Mangold et al. [68].

#### 4.5.2. Molecular Detection and Characterization of *Bartonella* spp.

A quantitative PCR targeting a fragment of the NADH ubiquinone oxireductase subunit G (*nuoG*) gene, using the primers F-Bart: 5′-CAATCTTCTTTTGCTTCACC-3′, R-Bart: 5′-TCAGGGCTTTATGTGAATAC-3′, and the hydrolysis probe FAM-5′-TTYGTCATTTGAACACG-3′[BHQ1]-3′ [69] was carried out for cat, dog and humans’ blood; ticks and colonies were suggestive of bartonellae for dog and human samples. The qPCR assays were performed for a final volume of 10 µL, containing 1 µL of DNA template, 0.6 μM of each primer and probe, 5 µL of Xpert Fast PROBE 2X Master Mix and ultra-pure sterilized water (Thermo Scientific). All samples were run in triplicate and the amplification conditions were 95 °C for 3 min followed by 40 cycles at 95 °C for 10 s and 52.8 °C for 30 s, using a CFX96 Thermal Cycler (BioRad™, Hercules, CA, USA). Standard curves were constructed using 10-fold serial dilutions of a gBLOCK^®^ (IDT-Integrated DNA Technologies) encompassing the 83 bp *B. henselae-nuoG* gene fragment. The number of copies was determined by the following formula: Xg/μL DNA/[fragment length in bp × 660]) × 6.022 × 10^23^ × plasmid copies/μL.

Positive samples for the *nuoG*-based qPCR gene were afterwards used in two cPCR assays. The citrate synthase gene (*gltA*) was targeted for amplification of a 767 bp fragment, using the primers 443F: 5′-GCTATFTCTGCATTCTATCA-3′ and 1210-R: 5′-GATCYTCAATCATTTCTTTCCA-3′ [70]. The 25 µL reactions consisted of 12.5 μL of NZYTaq II 2x Green Master Mix (NZYTech), 1 μM of each primer, 1 μL of DNA template and nuclease-free water. Cycling conditions were as follows: 95 °C for 5 min, 35 cycles of denaturation at 94 °C for 30 s, annealing at 54 °C for 30 s and extension at 72 °C for 1 min and final extension at 72 °C for 5 min. The second cPCR targets a 585 bp fragment of riboflavin synthase gene (*ribC*), using the primers BARTON-1: 5′-TAACCGATATTGGTTGTGTTGAAG-3′ and BARTON-2: 5′-TAAAGCTAGAAAGTCTGG CAACATAACG-3′ [71]. Reactions were made in the *gltA* cPCR assay and cycling conditions consisted of an initial denaturation at 95 °C for 5 min, 35 cycles consisting of denaturation at 94 °C for 30 s, annealing at 55 °C for 30 s and extension at 72 °C for 1 min, followed by a final extension at 72 °C for 5 min. Ultrapure water was used as a negative control. All assays were conducted in a T100™ Thermal Cycler (BioRad™, Hercules, CA, USA). PCR products were analyzed by horizontal electrophoresis in 1.5% agarose gel stained with Green Safe Premium (NZYTech). Positive amplicons for both cPCR targeting the genes *gltA* and *ribC* were purified using the NZYGelpure kit (NZYTech) and Sanger sequenced at StabVida (Lisbon, Portugal).

#### 4.5.3. Sequences and Phylogenetic Analysis

Sequences were analyzed and manually cured using Bioedit Sequence Alignment Editor [72], in order to enhance the quality level of the sequences. Apparent co-infections were evaluated by molecular cloning of the PCR product, using InsTAclone PCR Cloning Kit (Fermentas, Baden-Wurttemberg, Germany), and *E. coli* competent cells (NZYTech) according to standard cloning protocols. Briefly, the purified amplified product was ligated into the plasmid pTZ57R/T. Chemically competent *E. coli* cells (NZYtech) were transformed, plated in LB agar (NZYtech) and incubated at 37 °C, overnight. Positive colonies were subcultured overnight on 3 mL of LB medium (Liofilchem, Teramo, Italy) at 37 °C and under 200 rpm of shaking. Finally, plasmid purification was carried out using the NZY Miniprep kit (NZYtech) following the manufacturer’s recommendations, and 10 µL of purified samples together with 3 µL of M3 forward primer were sent to StabVida (Lisbon, Portugal), for sequencing. Sequences were manually trimmed and submitted to BLAST [73]. Edited sequences were grouped in four datasets (*B. henselae gltA*, *B. clarridgeiae gltA*, *B. henselae ribC* and *B. clarridgeiae ribC*), and a ClustalW multiple alignment was performed in Bioedit for each group of gene sequences. The number of haplotypes (h) and the haplotype diversity (Hd) was evaluated using the software DNA Sequence Polymorphism (DnaSP) v.6.12.03 [74]. The haplotypes were generated based on nucleotide sequences with gaps or missing data and excluding the invariable sites. MEGA v.6.06 software [75] was used to calculate the amount of nucleotide variation among the sequences using the *p*-distance model with 1000 bootstrap replications. For the phylogenetic trees’ construction, ClustalW multiple alignment was performed in Bioedit for each group of gene sequences (*gltA* and *ribC*), including representative sequences for each haplotype obtained in this study, together with other sequences selected from the GenBank (for *gltA*: *Bartonella acomydis* (AB444979.1), *Bartonella alsatica* (AF204273.1), *Bartonella birtlesii* (AF204272.1), *Bartonella vinsonii subp. vinsonii* (Z70015.1), *Bartonella vinsonii subsp. arupensis* (AF214557.1), *Bartonella jaculi* (AB444975.1), *Bartonella rattaustraliani* (EU111796.1), *Bartonella japonica* (AB242289.1), *Bartonella coopersplainsensis* (EU111803.1), *Bartonella quintana* (HQ014627.1), *Bartonella koehlerae* (AF176091.1), *Bartonella henselae* Human Australia (AJ439406.1), *Bartonella henselae* (L38987.1), *Bartonella henselae* Cat Flea Austria (MF374384.1), *Bartonella henselae* Cat Brazil (MH019304.1), *Bartonella henselae* Dog Chile (MG252490.1), *Bartonella doshiae* (AF207827.1), *Bartonella rattimassiliensis* (AY515124.1), *Bartonella queenslandensis* (EU111798.1), *Bartonella elizabethae* (Z70009.1), *Bartonella schoenbuchii* (AJ278184.1), *Bartonella chomelii* (AY254308.1), *Bartonella capreoli* (AF293392.1), *Bartonella rochalimae* (FN645459.1), *Bartonella clarridgeiae* Cat flea Chile (KY913636.1), *Bartonella clarridgeiae* Cat Brazil (MH019302.1), *Bartonella clarridgeiae* Cat Thailand (KX001761.1), *Bartonella clarridgeiae* (EU770616.1), *Bartonella clarridgeiae* strain 73 (FN645454.1), *Bartonella baciliformis* (DQ452947.1), *Bartonella tamiae* (DQ395177.1) and *Brucella abortus* (AE017223.1); and for *ribC: Bartonella henselae* Cat Brazil (HQ012583.1), *Bartonella henselae* fleas USA (AY953284.1), *Bartonella henselae* Cat Brazil (HM588661.1), *Bartonella koehlerae* (FJ832090.1), *Bartonella quintana* (AJ236917.1), *Bartonella washoensis subsp. cynomysii* (DQ825697.1), *Bartonella heixiaziensis* (KJ361664.1), *Bartonella vinsonii subsp. arupensis* (AY116631.1), *Bartonella vinsonii subsp. vinsonii* (AY116636.1), *Bartonella vinsonii subsp. berkhoffii* (AY116629.1), *Bartonella alsatica* (AY116630.1), *Bartonella fuyuanensis* (KJ361648.1), *Bartonella doshiae* (AY116627.1), *Bartonella sp.* Bat Kenya (HM363783.1), *Bartonella bovis riboflavin synthase* France (AY116637.1), *Bartonella chomelii* (AB290195.1), *Bartonella capreoli* (AB290194.1), *Bartonella schoenbuchensis* (AY116628.1), *Bartonella bacilliformis* (AJ236918.1), *Bartonella clarridgeiae* Cat China (EU571943.1), *Bartonella clarridgeiae* (AJ236916.1), *Bartonella clarridgeiae* Japan (AB292604.1), *Bartonella clarridgeiae* Cat Brazil (KR092386.1), *Bartonella ancashensis* (KP720649.1), *Candidatus Bartonella ancashi* Peru (KC886734.1), *Brucella melitensis* (CP008750.1)). No criteria regarding host or geographical localization were used to retrieve sequences from GenBank. Likelihood-mapping analyses were performed using the TREE-PUZZLE v5.3 program [76]. Only datasets with ≥90% resolved quartets were used for tree reconstruction. MEGA v.6.06 software was used to obtain the best model to describe the substitution pattern. A maximum-likelihood (ML) and a neighbor-joining (NJ) trees were constructed based on a bootstrapping method with 1000 replicates and Kimura 2-parameter model.

## 5. Conclusions

Bartonellosis is recognized as an emerging vector-borne disease that constitutes a threat to both animal and human health. Continuous and active surveillance to understand the epidemiological characteristics of bartonellosis is needed worldwide, including in Portugal. *Bartonella* spp. was not detected in ticks, dogs, and humans in the present study, but more studies need to be performed addressing this topic and putting emphasis on *Bartonella* spp. reservoirs and vectors. Regarding the dogs, surveys among endocarditis afflicted individuals can be carried out and similarly, studies conducted on humans should prioritize immune-compromised groups, as they are more prone to develop infections by *Bartonella* spp. About 20% of the cats sampled in Lisbon were positive for at least one *Bartonella* spp. and both *B. henselae* and *B. clarridgeiae* were identified in similar percentages. It is hoped that these results could bring attention to *Bartonella* sp. as an emerging pathogen in cats from Lisbon, and a potential threat for public health. As they are the main reservoir for *B. henselae*, preventive measurements should be implemented in order to avoid an outbreak of cat scratch disease in Lisbon.

## Figures and Tables

**Figure 1 pathogens-11-00749-f001:**
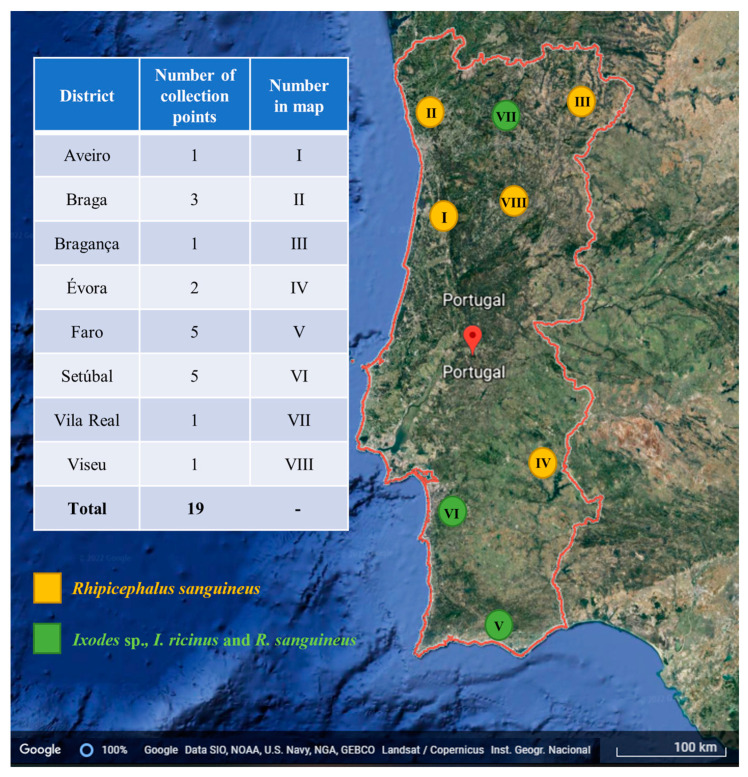
Ticks collected in Portugal mainland classified by district, with information of the number of collection points per district and whether *R. sanguineus* or both *R. sanguineus* and *Ixodes* sp. were identified. Image adapted from Google Earth v.7.3.2.5491.

**Figure 2 pathogens-11-00749-f002:**
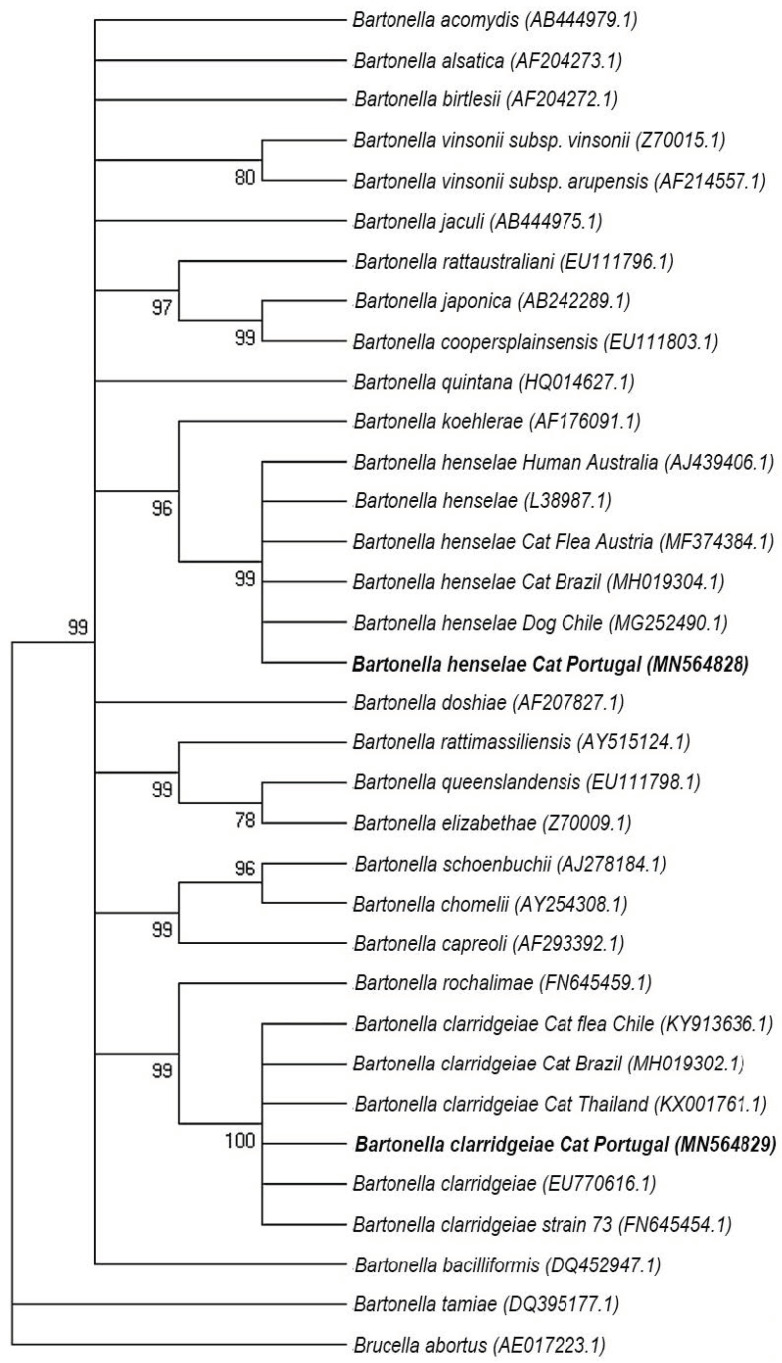
Phylogenetic analysis of *gltA* gene sequences (296 bp after alignment) based on the Maximum Likelihood method and model Kimura-2-Parameters. Numbers correspond to the support values for a bootstrap with 1000 repetitions, and only bootstraps >70% are presented. *Brucella abortus* was used as an outgroup.

**Figure 3 pathogens-11-00749-f003:**
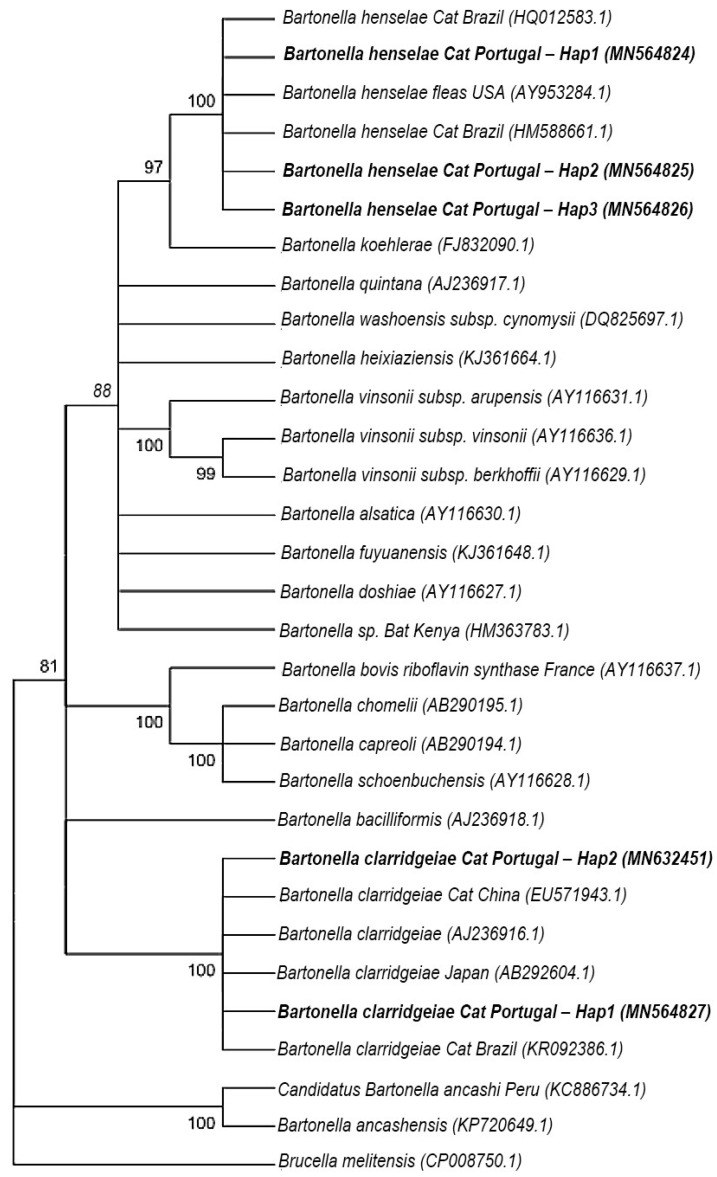
Phylogenetic analysis of *ribC* gene sequences (455 bp after alignment) based on the Maximum Likelihood method and model Kimura-2-Parameters. Numbers correspond to the support values for a bootstrap with 1000 repetitions, and only bootstraps >70% are presented. *Brucella melitensis* was used as an outgroup.

**Table 1 pathogens-11-00749-t001:** Samples information with regards to sex, age, and other characteristics.

	Number of Samples	Female	Male	Adult	Juvenile	Flea Infested
**Cats**	123	73 (58.8%)	51 (45.16%)	112 (91.05%)	11 (8.87%)	9 (7.25%)
**Dogs**	25	10 (43.47%)	15 (56.52%)	11 (47.82%)	12 (52.17%)	4 (17.39%)

**Table 2 pathogens-11-00749-t002:** Obtained *nuoG* qPCR reaction parameters for each type of sample.

Type of Samples	Efficiency (Average)	R^2^ (Average)	Slope (Average)	Quantification
**Ticks**	101.98%	0.996	−3.285	-
**Cats**	94.78%	0.994	−3.449	2.78 × 10 to 1.03 × 10^5^ copies/µL
**Dogs**	95.25%	0.996	−3.448	-
**Humans**	106.3%	0.678	−3.187	-

## Data Availability

The data presented are available within the article or in the case of sequences obtained, can be found at GenBank (https://www.ncbi.nlm.nih.gov/genbank/ accessed on 23 June 2022).

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
