# Peer review of "Molecular Survey of Bartonella Species in Stray Cats and Dogs, Humans, and Questing Ticks from Portugal"

_pathogens, 2022, doi:10.3390/pathogens11070749_

Round 1
Reviewer 1 Report
Material and Methods:
The point 4.5.1. PCR for DNA integrity confirmation should probably be named as in the work that the authors refer to, i.e. (26. Birkenheuer, A.J.; Levy, M.G.; Breitschwerdt, E.B. Development and Evaluation of a Seminested PCR for Detection and Differentiation of Babesia gibsoni (Asian Genotype) and B. canis DNA in Canine Blood Samples. J Clin Microbiol 2003, 41, 4172–4177, doi:10.1128/JCM.41.9.4172-4177.2003.) “As determined by amplification of GAPDH, no PCR inhibitors were detected in any of the DNA samples that were negative when our PCR test was used.” DNA integrity confirmation is usually checked by other methods, for example: measure absorbance from 230nm to 320nm to detect other possible contaminants.
Results:
The paragraph containing the sequencing results „For the dataset of B. henselae gltA (13 sequences), the number of sites found was 717, Pi = 0.000, Hd = 0.000 and only one haplotype was identified (MN564828). In the same way, the dataset of B. clarridgeiae gltA (11 sequences) reported a number of sites of 743, Pi = 0.000, Hd = 0.000 and only one haplotype was identified (MN564829). Furthermore, for the dataset of B. henselae ribC (14 sequences), the number of sites found was 530, Pi = 0.00081, Hd = 0.2747 and three haplotypes were identified: haplotype 1 (Hap1, MN564824), corresponding to 12 samples (2, 16, 27, 29, 37, 57, 62, 86, 90, 107, 108, and 119); haplotype 2 (Hap2 MN564825), corresponding to sample 95, and haplotype 3 (Hap3, MN564826), corresponding to sample 101. Similarly, regarding the B. clarridgeiae ribC group (11 sequences), the number of sites found were 521, Pi = 0.00035, Hd = 0.182 and two haplotypes were identified: Haplotype 1c (Hap1c, MN564827), corresponding to 10 samples (19, 22, 28, 33, 46, 52, 79, 80, 97, and 118), and Haplotype 2c, corresponding to sample 108 (Hap2c, MN632451).’’ should be presented before the results of the comparison of the obtained sequences with those available in GenBank.
Author Response
We sincerely appreciate the valuable comments and suggestions. The ms was changed according to the reviewer’s suggestions and we believe that the manuscript has improved and hope that you will now find it suitable for publication in Pathogens. Our point-by-point responses to comments are detailed below.
- Regarding the DNA integrity, all samples were checked on a NanoDrop ND-1000 spectrophotometer. This information was added to the manuscript and the remaining text concerning PCR Inhibition corrected.
- The paragraph was rearranged/moved as suggested.
Reviewer 2 Report
The manuscript (pathogens-1774488) submitted by Torrejón et al. consists in a molecular survey concerning Bartonella bacteria in several hosts, and in hard ticks regarding their possible role in the bacteria transmission.
First, without the line numeration, the manuscript reviewing is difficult since it can comprise where the authors should do the suggested alterations.
The competence of hard ticks as vectors of Bartonella is still a controversial issue, although in the discussion, the authors mention a few studies that presumably link ticks and bartonellosis, they are still highly circumstantial. Bartonella is common in wildlife, and it is natural that ticks can pick them up during a blood meal. The “Bartonella-positive” ticks can simply be ticks with residual Bartonella DNA that was acquired during a previous blood meal. This is very common – one can detect residual bacterial DNA and viral RNA post-feeding and even post molt.
So, I have some general questions that like the authors to address, namely:
· The studies regarding ticks and Bartonella are often developed in Ixodes Ricinus tick species, why did the authors decide to include also Rhipicephalus sanguineus sensu lato? Is it because it’s the “domestic animal” tick? A sentence justifying why the authors decide to also include this tick species could be added to the discussion section.
· Why didn’t the authors test the fleas that were on the sampled dogs and cats since it’s one of the proven vectors of this bacteria?
· Why weren’t the cultures also made with the cat’s blood samples? It could have been used as a “positive control” for the cultures.
Regarding each section of the manuscript:
Introduction:
What do you mean by mechanical infection due to contact with an animal reservoir?
Please introduce the complete gene´s name, since it’s the first time that there are mentioned in the manuscript.
Results:
The Portugal map is too small to see the regions where the ticks were collected.
Did any of the sampled animals have ticks?
Why do the qPCR parameters for human samples so low regarding R2?
The results regarding the sequence’s identity would be more comprehensive if presented as a table for each gene.
An outgroup should be included in the phylogenetic analysis.
The image quality of the phylogenetic tree is very poor.
Discussion:
When referring to all Portuguese population to justify that Bartonella infections in Portugal have a low incidence, please indicate the total population from Portugal, to better understand how low it is.
When referring to the several studies developed on ticks regarding the search of Bartonella, what were the tick’s species? Please add that information to the text.
Material and Methods:
Although the tick’s DNA extraction was performed following a previous protocol, please briefly describe that protocol.
4.5.2 Bartonella in italic like it is in the remaining manuscript
After “Thermo Scientific” please remove 10µL
4.5.3 – Please briefly explain how the co-infections molecular cloning were made.
How was the selection of the sequences for each haplotype made? Only European sequences? Worldwide? By host? By vector? All together? Please add that information in this section.
References:
Reference 78 can be removed.
Author Response
We sincerely appreciate the valuable comments and suggestions. The ms was revised according with the reviewer suggestions and we belive that it has improved. Hope that you will now find it suitable for publication in Pathogens. Our point-by-point responses to comments are detailed below.
- We apologize for the missing line numbering; we often complain about the same thing.
- Authors concur with the reviewer regarding the uncertainty on the role of ticks on Bartonella cycle. In light of that, we adjusted the phrase “Residual pathogen DNA, acquired during a previous blood meal, can be detected after tick molt and also Bartonella DNA within a tick does not imply that the tick might be able to transmit the bacteria during the course of blood feeding, therefore not conferring epidemiologic risk [73].
- Regarding the R. sanguineus, yes these ticks were screened due to its association with cats and dogs. In the discussion section we state that “Rhipicephalus sanguineus being the most prevalent tick in companion animals, has recently been used in similar studies [76,77] but evidence that this tick species can be the vector of Bartonella sp. is also still missing.”, however we added “These species were chosen due to their great importance for public health, I. ricinus the most medically important tick in Portugal (and Europe) and R. sanguineus the most prevalent tick in companion animals [69].”
- Fleas were only detected in a few number of cats and they were not collected at the time of the study since blood collections were made by the professionals at “Casa dos animais”
- We agree that cat blood samples could have been also cultured. In fact, we point that out on the ms: “(…) was not employed in the cat blood samples screened which would increase sensitivity and possibly yield in a higher number of positive samples [39].”. These samples were the first to be processed and we confirmed Bartonella infection with robust positive results after testing directly on the cats blood and didn´t stored sample for culturing.
- B. henselae transmission from cats to humans can occurdirectly by cat scratch or bite. We added this info in the sentence to clarify it.
- Corrected.
- The figure was automatically formatted in the submission. If necessary we will upload a better quality image.
- No ticks were detected on the animals sampled.
- The mean efficiency (E) of the qPCR reactions was 106.3% (100.4% to 112.2%), and the slope ranged from -3.313 (R2= 0.994) to -3.061 (R2=0.363).
- We have the tables done, but it seems repetitive since this info is in the text, however if the reviewer finds it crucial we will include it on manuscript.
- Brucella abortus and Brucella melitensis were used as outgroups for gltA and ribC, respectively.
- we are uploading the images in different format.
- Information regarding the Portuguese population was added “(…) (according to the national statistical institute, INE, in 2021, the Portuguese population was estimated in 10 344 802 individuals, https://censos.ine.pt)”
- The information regarding the tick species was added to the manuscript.
- A brief description was added as suggested.
- Bartonella is underlined here since all the topic is in italics.
- Corrected.
- A very brief description was added to the text. “Briefly, the purified amplified product was ligated into the plasmid pTZ57R/T. Chemically competent E. coli cells (NZYtech) were transformed, plated in LB agar (NZYtech) and incubated at 37°C, overnight.”
- We are not sure if we understand correctly the concern: The haplotypes were generated using our samples. “The number of haplotypes (h) and the haplotype diversity (Hd) was evaluated using the software DNA Sequence Polymorphism (DnaSP) v.6.12.03 [33]. The haplotypes were generated based on nucleotide sequences with gaps or missing data and excluding the invariable sites.”. We used sequences for the analysis from different hosts and countries. To calrify we added the information "No criteria regarding host or geographical localization was used to retrieve sequences from GenBank. "
- The reference was removed.